# Possible Mechanisms of the Interplay between Drugs and Mycotoxins—Is There a Possible Impact?

**DOI:** 10.3390/toxins14120873

**Published:** 2022-12-14

**Authors:** Orphélie Lootens, An Vermeulen, Siska Croubels, Sarah De Saeger, Jan Van Bocxlaer, Marthe De Boevre

**Affiliations:** 1Centre of Excellence in Mycotoxicology and Public Health, Department of Bioanalysis, Faculty of Pharmaceutical Sciences, Ghent University, Ottergemsesteenweg 460, 9000 Ghent, Belgium; 2Laboratory of Medical Biochemistry and Clinical Analysis, Department of Bioanalysis, Ghent University, Ottergemsesteenweg 460, 9000 Ghent, Belgium; 3MYTOX-SOUTH, International Thematic Network, Ghent University, Ottergemsesteenweg 460, 9000 Ghent, Belgium; 4Laboratory of Pharmacology and Toxicology, Department of Pathobiology, Pharmacology and Zoological Medicine, Faculty of Veterinary Medicine, Ghent University, Salisburylaan 133, 9820 Merelbeke, Belgium; 5Department of Food Technology, Faculty of Science, University of Johannesburg, Doornfontein Campus, P.O. Box 17011, Gauteng 2028, South Africa

**Keywords:** food–drug interaction, mycotoxins, CYP450-enzymes, human, pharmacotherapeutics, drugs, PBPK-modelling

## Abstract

Mycotoxin contamination is a global food safety issue leading to major public health concerns. Repeated exposure to multiple mycotoxins not only has repercussions on human health but could theoretically also lead to interactions with other xenobiotic substances—such as drugs—in the body by altering their pharmacokinetics and/or pharmacodynamics. The combined effects of chronic drug use and mycotoxin exposure need to be well understood in order to draw valid conclusions and, in due course, to develop guidelines. The aim of this review is to focus on food contaminants, more precisely on mycotoxins, and drugs. First, a description of relevant mycotoxins and their effects on human health and metabolism is presented. The potential for interactions of mycotoxins with drugs using in vitro and in vivo animal experiments is summarized. Predictive software tools for unraveling mycotoxin–drug interactions are proposed and future perspectives on this emerging topic are highlighted with a view to evaluate associated risks and to focus on precision medicine. In vitro and in vivo animal studies have shown that mycotoxins affect CYP450 enzyme activity. An impact from drugs on mycotoxins mediated via CYP450-enzymes is plausible; however, an impact of mycotoxins on drugs is less likely considering the much smaller dose exposure to mycotoxins. Drugs that are CYP450 perpetrators and/or substrates potentially influence the metabolism of mycotoxins, metabolized via these CYP450 enzymes. To date, very little research has been conducted on this matter. The only statistically sound reports describe mycotoxins as victims and drugs as perpetrators in interactions; however, more analysis on mycotoxin–drug interactions needs to be performed.

## 1. Introduction

Food contaminants comprise all unwanted dietary substances resulting from the applied cultivation conditions, production processes or environmental exposure. Certain contaminants pose a human health threat, such as toxic secondary metabolites produced by fungi, namely mycotoxins [1]. These toxins are detected in food crops such as maize, wheat, sorghum and peanuts [2]. They are produced by fungi as a self-protection mechanism during stressful conditions, and they are toxic to humans and animals, causing illness, and might even lead to death [3,4,5]. Fungi are able to produce multiple different mycotoxins, which lead to the existence of a great number of metabolites, exerting additive or even synergistic effects and causing (co)-morbidities and pathologies [5]. The exposure to these food contaminants is often chronic, depending on the geographical and climatic region of the world. High levels of contamination occur in regions where no strict regulations for mycotoxins are applied or where awareness is lacking, e.g., in low- and middle-income countries [6]. Research is imperative to study the impact of mycotoxin exposure on the pharmacokinetics (PK) and pharmacodynamics (PD) of drugs taken concomitantly and vice versa. For the first time, a summary of the current knowledge on this topic is reported.

## 2. Mycotoxins Are Linked to Human Diseases

### 2.1. General Overview of Mycotoxins and Their Toxic Effects

In 1960, an outbreak of aflatoxicosis in turkeys occurred due to toxic groundnut meal from Brazil. More than 100,000 turkeys died (‘Turkey-X-disease’). From this moment onwards, research has been performed to understand the mycotoxins and their effects; and to apply stringent regulatory standards [7]. *Fusarium*, *Alternaria*, *Penicillium*, *Claviceps* and *Aspergillus* are the most important fungi that can produce several different mycotoxins. The most abundant mycotoxins in food crops are aflatoxin B1 (AFB1), aflatoxin B2 (AFB2), aflatoxin G1 (AFG1), aflatoxin G2 (AFG2), alternariol (AOH), alternariol methyl ether (AME), ochratoxin A (OTA), deoxynivalenol (DON), zearalenone (ZEN), fumonisin B1 (FB1), fumonisin B2 (FB2), fumonisin B3 (FB3), T-2 toxin (T-2), HT-2 toxin (HT-2), nivalenol (NIV), 3-acetyldeoxynivalenol (3-ADON), 15-acetyldeoxynivalenol (15-ADON), diacetoxyscirpenol (DAS), fusarenon-X (F-X), neosolaniol (NEO), roquefortine C (ROQ-C) and sterigmatocystin (STERIG) [2]. Aflatoxins (AFs) are present in maize, rice, spices, nuts, seeds and soybeans [8]. OTA is often present when the relative humidity is high, and occurs in different food types, e.g., wheat, berries, grapes, coffee and certain spices. DON is found in wheat grain, maize and barley. ZEN is present in maize and is found in maize oil, cereal grain and other products. Fumonisins (FBs) are detected in maize and therefore in maize products e.g., popcorn and nachos. T-2 is found in cereals, and the same holds for HT-2, both belonging to the trichothecene family [9]. T-2 and HT-2 can induce oxidative stress and block DNA-, RNA- and protein synthesis, which likewise applies to other trichothecenes e.g., DON [10]. Whereas not all of the fungal metabolites are dangerous to health, as some are applied as growth promoters, antibiotics or other drugs, [3] mycotoxins’ exposure may generally lead to a variety of toxic effects as summarized in Table 1.

Mycotoxicosis, which is referred to as poisoning by mycotoxins, is due to acute or chronic exposure to these compounds. Associating or relating health effects to a specific type of exposure is a challenge, since chronic exposure is a daily fact. In general, a toxic response is caused by acute exposure, while diseases such as cancer are due to chronic exposure [20]. Mycotoxicosis depends on the causing mycotoxin, age, dietary status, predisposing genetic factors and the duration of the exposure. Other factors, such as the general health status of the person and existing comorbidities, play a role in the outcome of the mycotoxicosis [21]. Reports on the toxicological effects, toxicokinetics and occurrence data of mycotoxins have been published and are detailed in comprehensive reviews [1,6,22,23].

### 2.2. Mycotoxins: A Global Health Problem

Mycotoxins are a worldwide problem, but some regions in the world are more affected than others due to the absence of strict regulatory standards, shortage of resources, shortcomings in knowledge and differences in climatological conditions. Maize and rice are commonly used food products, known to be prone to *Aspergillus* and *Fusarium* infections, consequently leading to AFs and FBs exposure. Fungal invasion is favored in conditions that are not in line with the known critical safe storage limits, e.g., when grain moisture exceeds 14% and when the temperature is between 25–37 °C [24]. Differences in climatic conditions between regions have an impact on the growth of fungi and subsequent mycotoxin production. Humid conditions increase the fungal invasion but periods of stress (e.g., drought) induce the mycotoxin production of, e.g., AFs and FBs. It is suggested [25] that the mycotoxins patulin (PAT) and OTA also becomes more prevalent in warmer climatic conditions. Climatic differences in regions induce differences in mycotoxin exposure, but global climate change induces alterations in mycotoxin exposure [24]. Sub-Saharan Africa, e.g., is heavily affected by mycotoxins due to climatic conditions, moreover aggravated due to the fact that subsistence farmers have scarce to no knowledge about the existence of mycotoxins and their accompanying hazards. Food is often stored in a suboptimal way, making it sensitive to fungal invasion. The most effective strategy to prevent the presence of mycotoxins is to protect food crops from fungal invasion. Chemical decontamination is prohibited since the health effects have not been studied. Primary prevention has to be implemented before fungi infest the crops. Fungi-resistant plant varieties and fungicides are optional, while the moisture content of plant seeds can be lowered during storage. In general, all storage conditions must be optimized regarding temperature, moisture content and light exposure [23].

In 2004, in Kenya a major outbreak of mycotoxicosis, caused by AFs, occurred and was accountable for 125 deaths [26]. In 2005 and 2006, smaller outbreaks were reported in the same regions with a total of 53 confirmed deaths. The number of deaths due to mycotoxins is not conclusive since not all diseases and deaths were monitored. Villages are often remote in these regions and a robust reporting system is missing [22]. In 2016, in Tanzania, many children and elderly died due to high levels of AFs in the home-grown food commodities with ensuing high levels of aflatoxin adducts with serum albumin. Aflatoxicosis was put forward as the cause of these deaths [27]. More recently in 2019, a suspected aflatoxicosis outbreak was reported in Tanzania where 8 of the 53 cases died [28]. A study in Sierra Leone observed that children are chronically exposed to both AFs and OTA, but that the exposure is higher during dry seasons compared to rainy seasons [29]. Another study in the Kathmandu Valley in Nepal over four different seasons during 2007–2008, indicated that drier periods lead to higher levels of mycotoxin contamination [30]. Positive correlations have been found between hepatocellular carcinoma (HCC) and a high dietary consumption of AFs in Africa [31,32]. An estimation was made of HCC cases due to AFs on a global scale, and set between 105,000 and 142,000 cases annually [33]. Furthermore, illnesses, deaths and disease-attributable-life-years-lost (DALY) due to HCC associated with AFs were estimated at 56,247.63 in Tanzania [34]. However, not only AFs, but also multiple other mycotoxins are associated with the onset of HCC and colorectal cancer. In addition, other mycotoxin-related health issues were reported in Africa, e.g., nephropathy caused by OTA in Ivory Coast [35] and Tunisia [36], stunting in Benin, Togo and Tanzania due to AFs and FBs [37], immunodeficiency [38], liver diseases [39], anemia [40] and infertility [41]. In 1989, a human mycotoxicosis outbreak occurred in the subtropical Kashmir Valley in India, which was caused by the consumption of wheat products that were contaminated with DON, leading to nausea, vomiting, diarrhea, abdominal pain and fever [42]. However, not only human health problems occur. Animals are also sensitive to mycotoxin exposure. In the United States of America (USA) in 2005, seventy-five dogs died because of pet food contaminated with AFs [43]. In Argentina, an acute aflatoxicosis case was observed in a chinchilla farm, where 200 animals died [44]. Acute mycotoxicosis cases were all caused by AFs that were present in maize. When maize becomes an important food/feed staple, the risk for mycotoxicosis rises. In 2021, analysis of finished feed samples in North America by Biomin (DSM Group) was not promising in terms of mycotoxin contamination. The verdict was that DON was present in 81 percent of all analyzed samples (*n* = 16,164) with an average contamination of 846 µg/kg. An average concentration of 1840 µg/kg was detected for feed samples positive for FBs. Furthermore, it was stated that AFs and ZEN contamination was increased compared to 2020 [45]. A similar phenomenon was observed in Europe, where an increase for DON, ZEN and FBs in animal feed was observed in 2021. These findings are probably due to a higher atmospheric temperature, water stress and increased CO_2_-levels, all linked to climate change [46]. In addition, hot spots of high DON contamination were found in wheat and barley samples in Europe [47].

## 3. Mycotoxin–Drug Interactions Are an Emerging Research Topic

### 3.1. Food and Pharmacology: Food–Drug Interactions

Drugs interact with the diseased human body with the aim to bring benefit in a clinical context. However, a clinical condition, food, other drugs or a certain lifestyle may influence both the PK and PD of drugs. These influences may lead to a higher or lower effect of the drug, potentially leading to a changed clinical outcome. When drugs and food components are substrates for a certain enzyme, e.g., cytochrome P450 (CYP450) enzymes, a competitive or non-competitive interaction may occur when taken simultaneously. These interactions can be reversible or non-reversible, inhibitory or inducing [48]. The drug and food component may directly compete for metabolism via an enzyme, implying a competitive interaction. They may also bind to different sites of the enzyme, inducing a non-competitive interaction between the drug and food components through influencing a three-dimensional configuration of the enzyme’s active center.

Food–drug interactions are an established area of research: nutrients can show a lower absorption due to interferences with concomitantly taken drugs. However, more importantly, at the level of metabolism and excretion there are interactions possible with potentially clinically relevant effects such as toxicity or ineffectiveness of the drug [49]. Drugs that interact with the metabolism of other exogenous compounds could also have a similar impact on the exposure as well as the effects of mycotoxins. Therefore, it is of utmost importance to thoroughly investigate the potential for mycotoxin–drug interactions, especially in a context where mycotoxin exposure is commonplace and highly prevalent, such as in low- and middle-income countries. Drugs of interest are, in this case, drugs that are commonly consumed in the areas where mycotoxin contamination is high (e.g., human immunodeficiency virus (HIV)-blockers in South Africa).

### 3.2. Mycotoxin Metabolism through The CYP450 Complex

Mycotoxins—as with every xenobiotic—undergo absorption, distribution, metabolism and excretion in the human body. Metabolism occurs in different parts of the human body, but mainly by hepatic enzymes, and can be divided in phase I (oxidation, hydrolysis and reduction) [50], and phase II (conjugation) reactions [51]. Metabolism is a way to facilitate excretion from the body, among others, but it might lead to both detoxification and bioactivation of the parent compound. Considering all enzymes, the CYP450 complex is an important group involved in phase I metabolism, mainly found in the liver and gut. It is the most relevant enzyme family to consider, in view of its predominant role in drug metabolism, since it is involved in approximately 80 percent of all drug metabolism processes, as well as in the metabolism of endogenic compounds [52]. Variability in expression and activity of CYP450 enzymes is known to occur in humans and animals, mostly due to genetic polymorphisms [53]. In addition, factors such as age, gender, and health status have an impact on CYP450 enzyme expression and activity [54]. Different types of interactions are possible, potentially leading to higher or lower effects of the administered drugs or a less or more toxic effect of the mycotoxins one is exposed to. The most encountered mycotoxins and their metabolism are discussed below, i.e., AFs and *Fusarium* toxins.

#### 3.2.1. Aflatoxins

Metabolism of AFs via CYP450 results in different metabolites such as aflatoxin M1 (AFM1), aflatoxin Q1 (AFQ1), aflatoxin-exo-8,9-epoxide and aflatoxin-endo-8,9-epoxide (AFBO) [55]. AFQ1 and AFM1 are annotated as detoxified metabolites, whereas AFBO is considered a bioactivated metabolite, exerting carcinogenicity. CYP1A2 and CYP3A4 are the most important enzymes involved in the metabolism of AFB1 but also CYP3A5 and CYP3A7 play a role (Table 2) [55]. The intrinsic toxicity of AFB1 does not imply mutagenicity, though the epoxidation products of AFB1, AFBO, will form DNA-adducts and is therefore carcinogenic to humans. The DNA-adduct that will be formed is 8,9-dihydro-8-N7-guanyl-9-hydroxy aflatoxin (AFB1-N7-Gua), which is converted into AFB1-formamidopyrimidine (AFB1-FAPY), both leading to mutations [56]. Metabolism not only takes place in the liver and gut, but also in the respiratory tract after inhalation of, e.g., infected maize dust, where CYP2A13 will activate AFB1 to form the N7-guanine-adduct [57]. Large variations in AF metabolism have been observed both between and within species [58]. Age, gender and other factors are known to have an impact on the metabolism, making the extrapolation between species difficult [53].

#### 3.2.2. Fusarium Toxins

Via phase II metabolism, DON is converted to DON-3-sulfate in the intestine, liver and kidney in poultry, and is rapidly eliminated from the body via excreta [59]. Animals produce more DON-metabolites compared to humans; only pigs show similar DON toxicokinetics and metabolism [60]. In the human body, a part of the consumed DON-fraction is metabolized to DON-glucuronides in the liver and gut by phase II uridine diphosphate-glucuronosyltransferase enzymes (UGT) [61]. More than 75 percent of the DON found in urine appeared to be in a glucuronidated form. Warth et al. (2013) [62] and Sayyari et al. (2018) [63] concluded that CYP3A4 was not involved in the metabolism of DON [62,63]. Moreover, none of the CYP450 enzymes are involved in the metabolism of DON [58]. DON-15-glucuronide (DON-15-Glc) is the most abundant DON metabolite found in human urine [61]. Other DON-metabolites such as iso-DON-8-glucuronide (iso-DON-8-Glc), iso-DON-3-glucuronide (iso-DON-3-Glc) and de-epoxy deoxynivalenol-3-glucuronide (DOM-3-Glc) have been identified in urine of humans. Those metabolites were also found in pig urine and plasma, but in very low concentrations, whereas DON-3-Glc and DON-15-Glc appear to be the dominating metabolites [61,62,63].

With the same molecular backbone as DON, T-2 belongs to the trichothecene family. T-2 gets rapidly metabolized into a number of metabolites, mainly phase I metabolites, e.g., 3′-OH-T-2, 3′-OH-HT-2, HT-2, T-2 triol and glucuronides thereof (phase II) [58]. Another study that focused on ruminants and non-ruminants showed that the initial step in T-2 biotransformation is deacetylation leading to the formation of HT-2 [64]. Carboxylesterases are the most important enzymes for the metabolism of T-2, followed by CYP3A4, CYP2E1, CYP1A2, CYP2B6, CYP2D6 and CYP2C19 of the CYP450 complex, as shown in in vitro experiments in human liver microsomes [65].

Via phase I metabolism ZEN is converted to α-zearalenol (α-ZEL) and β-zearalenol (β-ZEL) via the enzymes α-hydroxysteroid dehydrogenase and β-hydroxysteroid dehydrogenase, respectively, as well as via CYP-enzymes, and phase II metabolism to glucuronides thereof. The intestine is the most important site of ZEN metabolism after oral ingestion. In vitro tests with human CYP450 enzymes demonstrated the oxidation of ZEN via CYP2C8, CYP3A4 and CYP3A5. An inhibition of CYP2C and CYP3A was observed due to the presence of ZEN [66]. Another study discovered that CYP3A4 and CYP1A2 are responsible for the hydroxylation of ZEN into catechol metabolites. This study proved that aromatic hydroxylation of ZEN is the main metabolic pathway in vitro [67].

Unlike AFB1, FBs do not get extensively metabolized. In rats, the majority of consumed FBs were excreted in their unchanged form. A study has also been performed in monkeys where hydrolysis products of FB1 have been observed in the feces [68]. FB1 causes a small increase in CYP1A activity and expression [69]. Another study showed that there is no phase I metabolism of FB1, but that the intestinal microbiota in pigs and poultry can hydrolyze FB1 [70,71].

Using two in vitro systems, namely human liver microsomes (HLM) and CYP3A4-containing nanodiscs (ND), it was demonstrated that ENN B1 is mainly metabolized by CYP3A4 and CYP3A5 [72]. This study identified 11 metabolites of ENN B1. In addition, DON was studied, clarifying that it was not metabolized by CYP3A4, though it did interact with the metabolism of ENN B1, since DON decreased its metabolic rate. This finding indicated that DON may cause a non-competitive inhibition of CYP3A4, leading to a slower metabolism of ENN B1. A comparative in vitro and in vivo metabolism study of ENN B1 in pigs showed that it was extensively metabolized in pig liver microsomes, which was confirmed in vivo after both intravenous and oral administration. The main metabolites were observed in higher levels after oral administration as compared to intravenous administration. This indicates that pre-systemic metabolism contributed to ENN B1′s metabolism when taken per os [72]. A different study was performed, where the metabolism of ENN B1 was confirmed in microsomes of rats, dogs and humans. Additionally, phenotyping of the CYP450 enzymes was performed using chemical inhibitors, which are selective for specific human enzymes. CYP3A4 was the main enzyme involved in the metabolism, followed by CYP1A2 and CYP2C19. This confirmed the results of the previously discussed studies where CYP3A4 was pinpointed to be the principal enzyme involved in the metabolism of ENN B1 [73]. As shown in Table 2, the hepatic biotransformation of certain mycotoxins is well understood. It is clear that chronic exposure to mycotoxins may potentially lead to an altered biotransformation of xenobiotics or vice versa, such as drugs, in the liver and/or the gut through interaction at the level of the biotransformation enzymes.

Certain mycotoxins are metabolized via CYP450 enzymes; all drugs that are known CYP450 perpetrators and/or substrates will have an impact on the metabolization of mycotoxins, wherein the same CYP450 enzymes are involved. The formation of less (inhibition) or more (induction) metabolites is crucial for mycotoxins and the exertion of their toxicity, taking into account the biochemical activation of some mycotoxins (e.g., AFB1 is metabolized into the more toxic AFBO via CYP450 enzymes). Figure 1 pinpoints the different pathways of ENN B1, T-2, ZEN and AFB1, as a CYP450 substrate or CYP450 perpetrator, to their metabolites. Drugs that are strong CYP3A4-inducers (e.g., carbamazepine, an anti-epileptic drug) or strong CYP3A4-inhibitors (e.g., lopinavir, a human immunodeficiency blocker) will potentially impact the metabolism of the mentioned CYP3A4-substrate mycotoxins. Inducers will lead to more metabolite formation, which is satisfactory in case of detoxification, but not in case of bioactivation (e.g., AFBO formation). Inhibitors will lead to less metabolite formation.

## 4. Mycotoxin–Drug Interactions: In Vitro and In Vivo Studies

### 4.1. General Overview

A report by Anyanwu et al. (2004) discussed the interaction between mycotoxins and antifungal drugs, administered to treat indoor chronic toxigenic mold exposures. Exposure to mycotoxins mainly happens through contaminated food consumption; however, another possible route is via dermal absorption or inhalation [74]. Mycotoxins are possibly released via sporulation of indoor fungi, which can be absorbed through the skin and airways [75]. Antifungal drugs are mostly derived from azoles, allylamines, antifungal polyenes and mycotoxins. The effectiveness of certain antifungal drugs probably depends on the mycotoxin prevalence in the human body [74]. Research indicated that when a patient is exposed to a mycotoxin possessing an azole structure, and is treated with an azole antifungal drug, the clinical response will not be fully effective. At first, the patient’s health will improve slightly, but it will recede since the drug will interact with the present mycotoxins in the patient’s serum instead of interacting with the targeted fungi [74]. This research especially considered the structural interactions between mycotoxins and antifungal drugs, since the latter are often structurally related. Possible impact on drug-metabolizing enzymes from mycotoxins have only been investigated in a few studies. Mycotoxins, especially AFB1 and T-2, have an inhibitory effect on certain enzymes in animals [76]. In some cases, there was an increase in activity and protein expression of CYP450 enzymes and glutathione transferase, which was the case for AFs, FBs and DON. Mycotoxins can induce/inhibit enzyme expression and they might interact with other xenobiotics at the enzyme level. Of note, the studies have thus far been performed in rodents, pigs, rabbits and/or poultry, but not yet in humans [76]. The few studies that have been performed on this topic strengthen the hypothesis of possible interactions with drug metabolism when animals and humans are (chronically) exposed to high concentrations of mycotoxins.

Next to direct induction or inhibition of enzymes by mycotoxins, ZEN has been shown to have an influence on the human pregnane X receptor (PXR), which is responsible for the regulation of CYP450 enzyme genes, among others. PXR also regulates the transcriptional regulation of phase II enzymes and ABC efflux transporters [77]. As further described below, no statistically relevant interaction has been elucidated for ZEN via this mechanism, but it is a hypothetical mechanism that might be involved in mycotoxin-drug interactions. Table 3 is a summary on the current knowledge of the effect of AFB1, DON, T-2, ZEN, FB1 and ENN B1 on CYP450 enzymes. This summary consists of in vitro work and animal in vivo research so it might not be representative for human interactions.

Next to the impact on CYP450 enzymes, exposure (acute or chronic) needs to be taken into consideration. AFB1 levels of 355 µg/kg have been reported in homegrown maize kernels [26]. In one case, levels of 16,505 µg/kg AFB1 were found in peanut butter in 2001 that was given to school children in South Africa [85]. It is noteworthy that it is more likely that drugs will have an impact on the PK of mycotoxins than mycotoxins will have an impact on drug disposition, considering the concentrations of exposure of the mycotoxins versus drugs (mg levels). Table 4 gives an overview of the mycotoxin–drug interactions that are described in the next subchapters.

### 4.2. Aflatoxins

Research in turkeys, a species highly sensitive to aflatoxicosis, revealed an AFB1 bioactivation via CYP1A1, but not via CYP1A2. The enzyme CYP1A5 was also involved in the biotransformation of AFB1 in turkeys. Interestingly, this CYP1A5 enzyme shows a large similarity in amino acid sequence with human CYP1A1 and CYP1A2 [92]. Additionally, in chickens, Watanabe et al. (2013) demonstrated that CYP1A5 genes are the orthologues of human CYP1A1 and CYP1A2 [93]. A study has been performed by Guarisco et al. (2008) in turkey microsomes to investigate the interaction between AFB1 and butylated hydroxytoluene (BHT), an antioxidant often added to animal feed. In rodents, it was observed that BHT protected the animals against carcinogenesis caused by AFB1 [86]. The study by Guarisco et al. (2008) showed that BHT protects turkeys against aflatoxicosis by inhibiting the bioactivation into AFBO, a carcinogenic metabolite of AFB1. Different CYP450 enzymes such as CYP1A2 and CYP3A4 are involved in the metabolism of AFB1. Methoxyresorufin O-demethylase (MROD) activity, a prototype for CYP1A2 activity, was decreased in turkeys fed with BHT. Nifedipine oxidation, indicative for CYP3A4 activity, was unexpectedly increased in turkeys fed with BHT (4000 mg/kg). In vitro, both CYP1A2 and CYP3A4 activities were decreased in liver microsomes treated with BHT. A side effect of long-term BHT consumption was a hydropic degeneration of livers, that appeared to be reversible and did not cause permanent damage. In turkeys, the activation of CYP3A4 by BHT led to a decreased in vivo formation of AFBO, implying that CYP3A4 is not involved in the biotransformation of AFB1 in turkeys [86]. This is in contrast with the metabolism of AFB1 in humans, where CYP3A4 is involved, but only in the presence of relatively high AFB1 concentrations [94]. In ducks, CYP3A4 involvement was proven in the metabolism of AFB1. Diaz and Murcia (2019) demonstrated a discrepancy in AFB1 sensitivity (ranked from resistant to highly sensitive) between chickens, quails, turkeys and ducks [95]. The explanation for the high sensitivity of ducks to AFB1 has to be found in a different metabolism pathway. Ducks produce large quantities of AFB1-dihydrodiol (AFB1-dhd), which is produced from the AFBO metabolite and might be the cause of ducks being the only poultry species that develop HCC [95]. A study performed by Corcueral et al. (2014) investigated the combined genotoxicity of AFB1 and OTA in rats, after the simultaneous application of the in vivo Comet Assay and the micronucleus test (MN) [96]. An alteration in the toxic effect of AFB1 was observed when simultaneously applied with OTA, possibly due to metabolic factors. Co-administration of AFB1 and OTA led to decreased levels of AFB1 in plasma and tissues, but increased levels of OTA. Hence, OTA protects the body from AFB1 genotoxicity. Further research is needed, but it is hypothesized that OTA, which is more present in plasma and tissues when administered simultaneously with AFB1, forces AFB1 into other metabolic pathways that do not or only to a lower extent lead to AFBO formation [96]. The effect of oltipraz (OPZ), a synthetic dithiolthione and chemopreventive and anti-angiogenic agent, on the biotransformation of AFB1 in rats [88], marmoset monkeys [89] and in human hepatocytes [90] was studied. All studies came to the conclusion that OPZ had a chemoprotective effect since it lowered the biotransformation of AFB1 into its carcinogenic AFB1-8,9-epoxide, AFBO. A decrease in DNA-adducts was observed and this might be partially explained by the inhibition of CYP450 enzymes that are involved in the bioactivation of AFB1 into AFBO. OPZ was also used in a human in vivo chemoprevention trial in the Qidong population in China. This study was performed by Kensler et al. (1998) in 1995 and 234 adults were enrolled [87]. The Qidong region has a high prevalence of HCC, partially explained by the consumption of aflatoxin contaminated food commodities. The trial was double-blind, randomized and placebo controlled. Participants were divided into three groups i.e., a placebo group, a group that received 125 mg OPZ once daily and a group that received 500 mg of OPZ once weekly, all during the study period of 8 weeks. Unfortunately, no statistically significant differences were observed between the three groups. Noteworthy, there was a triphasic effect observed in the 500 mg OPZ group, where no effect was observed in the first month, a decrease in AFB1-adduct formation occurred in the second month of OPZ intake and in the first month post-intervention [87]. Recently, PK parameters of AFB1 were unraveled by Lootens et al., (2022) giving more insight in what happens with AFB1 in the human body [97]. It is clear that more research needs to be performed both in vitro and in vivo to have a better insight in possible interactions between mycotoxins and other compounds, such as drugs. Interactions need to be unveiled since they might lead to less or even more toxic effects of the mycotoxins but could also lead to less or more toxic effects of the interacting compound.

### 4.3. Fusarium Toxins

Research performed by Schelstraete et al., (2019) initially studied the in vitro effects of ZEN, T-2, FB1 and DON on six porcine CYP450 enzymes. Based on the in vitro inhibition experiments, indicating a mycotoxin—CYP3A substrate interaction, the focus of the in vivo study was on the impact of subacute exposure to T-2 and ZEN on the PK properties of midazolam (a CYP3A model substrate) in a porcine animal model [91]. Midazolam (MDZ) was used as a CYP3A probe substrate since it undergoes substantial first-pass metabolism in the human liver and intestine. A potent inhibition of the metabolism of MDZ mediated via CYP3A by T-2 and ZEN in pigs was demonstrated [91]. DON did not inhibit any of the studied CYP-enzymes for more than 20 percent and was not further evaluated in the in vivo study. FB1 showed a reduction in the activity of CYP2E, CYP2A and CYP1A but this was considered to be due to the inhibition of CYP2A19 (involved in the biotransformation of phenacetin and chlorzoxazone in pigs, not in humans). Therefore, FB1 was not further investigated neither. The study by Schelstraete et al. (2019) revealed that T-2 and ZEN have an impact on the PK of MDZ by their effect on the intestines rather than the liver [91]. The results showed a potential inhibition of intestinal CYP3A by ZEN and a possible disruption of the intestinal barrier function by T-2, due to the cytotoxic effects of the latter compound. The effect of T-2 and ZEN on the metabolism of MDZ was only statistically significant for the differences in elimination constant (K_e_) estimates. This implies that larger studies need to be performed in order to confirm the findings. The main conclusion of this study was the potential for in vivo interaction of T-2 and ZEN present in a variety of foodstuffs, at worse-case scenario levels of mycotoxin exposure, with the metabolism of drugs handled by the same CYP-enzymes [91].

### 4.4. Physiologically-Based Pharmacokinetic Modelling to Predict Mycotoxin-Drug Interactions

The implementation of physiologically-based pharmacokinetic (PBPK) modelling allows outcome predictions for certain clinical or pathophysiological situations based on in vitro and/or in vivo data, instead of a dedicated (clinical) trial. For a general comprehensive review on PBPK modelling, the authors refer to Zhuang and Lu, 2016 [98]. PBPK models represent the physiological structure of an animal or human as consisting of certain compartments. Each compartment represents organs of the body with similar characteristics and is attached to the circulating blood system. Assumptions are made for each compartment, e.g., the drug or mycotoxin undergoes perfusion-rate-limited or permeability-rate-limited distribution to tissues. In this way, the physiology of the body and the fate of the pharmacotherapeutic or mycotoxin regarding, e.g., elimination are modelled, and by taking this knowledge into account, the pharmacokinetic behavior of chemical compounds is predicted as if it was administered in vivo. The applied parameters, e.g., metabolic clearance, used to construct such a model are obtained from in vitro and/or in vivo experiments. This tool is used in the pharmaceutical industry, both in pharmacotherapeutic discovery and development, as well as in academical research settings. These PBPK models are applied to represent animal species such as mice, rats, monkeys and dogs, but also humans. Within the human population, variations are/can be introduced regarding parameters such as ethnic background, age, gender and health status. Furthermore, drug–drug interactions (DDI) are frequently simulated using the described approach. In the particular case of mycotoxin–drug interactions, simulations of co-administration of mycotoxins with certain drugs can be performed. Since DDI-studies—or in this case, food contaminant-drug interactions studies—are often not ethically justified, a fine-tuned, qualified and well-established PBPK-model could present us with a reliable tool to predict non-studied clinical in vivo conditions. Correctly determined input parameters and compound characteristics have to be used to obtain a reliable prediction. Additionally, correct assumptions have to be made and the predicted values need to be thoroughly validated (e.g., through in vivo animal testing). Nevertheless, PBPK modeling has established itself as a promising and helpful tool that enables to reduce animal testing as much as possible and reduces the need for additional human in vivo testing when a robust, fine-tuned PBPK model is established [99]. Previously, a physiologically-based toxicokinetic (PBTK) model for T-2 in chicken and a human PBPK model for DON were built [100,101]. For AFB1, a PBK model was built based on in vitro–in silico testing [102]. These models were developed to simulate exposure to a certain mycotoxin and not yet to simulate mycotoxin–drug interactions.

## 5. Future Perspectives

Food in general may affect the response of the body to drugs, through altered absorption, distribution, metabolism and/or excretion. Additionally, drugs might also affect food absorption and metabolism. Parameters such as gastrointestinal motility might be altered due to drugs, food or food contaminant consumption [103]. Therefore, it is of utmost importance to elucidate all the processes that may have an impact in order to understand clinical outcomes and potential interactions. Possible interactions between drug-metabolizing enzymes and mycotoxins have only been investigated in a few animal in vivo studies [86,87,89,91,92,93,94,95] while a few other studies focused on the impact of mycotoxins on in vitro enzyme activity [88,90,96]. The inhibitory effects of AFs, FBs and DON on certain enzymes reported by Galtier et al. (2008) were highlighted already [76]. Mycotoxin–drug interactions are a novel research domain where in vitro experiments and in vivo animal research complemented with well-established PBPK-models will allow extrapolations to the human in vivo situation, limiting the need for ethically constrained in vivo experiments.

## 6. Conclusions

The biotransformation of mycotoxins needs to be thoroughly investigated, with the aim to unravel possible interactions with concomitantly taken drugs. If relevant interactions are found, the findings may, in due course, be used to set up guidelines regarding the possible concomitant intake of drugs and mycotoxins. So far, only a few reports discuss this research topic, particularly focusing on animal species. However, extrapolation from other species to humans is not straightforward. The use of PBPK models might predict reliable outcomes for non-studied in vivo situations, as an ethical dilemma is present for setting up controlled and well-designed studies in humans involving high exposure to mycotoxins and drugs. Focus needs to be set on the frequently occurring mycotoxins in targeted regions where specific drugs are administered. Since some mycotoxins are metabolized by CYP450 enzymes, it is plausible that drugs, which are known CYP450 perpetrators will impact the metabolism of CYP450 metabolized mycotoxins, changing the PK and PD of the mycotoxins. As shown in literature, mycotoxins have an impact on CYP450 enzymes; this might lead to interactions. Various in vivo animal experiments show differences in impact of mycotoxins on CYP450, indicating that it is species-specific. Considering the dose differences between mycotoxins and drugs, it is more likely that drugs will impact mycotoxin metabolism than vice versa. The authors state that mycotoxin–drug interactions in humans are an emerging research topic. To the authors’ knowledge, mycotoxin–drug interaction research is not an active investigation field to date. Nonetheless, eventually confirmed interactions might lead to future recommendations on drug-dose adaptations or a switch-over to other drugs and where interdisciplinary research fields such as food safety, DDI-research, PBPK-modelling and pharmaceutical research will be complimentarily joined.

## Figures and Tables

**Figure 1 toxins-14-00873-f001:**
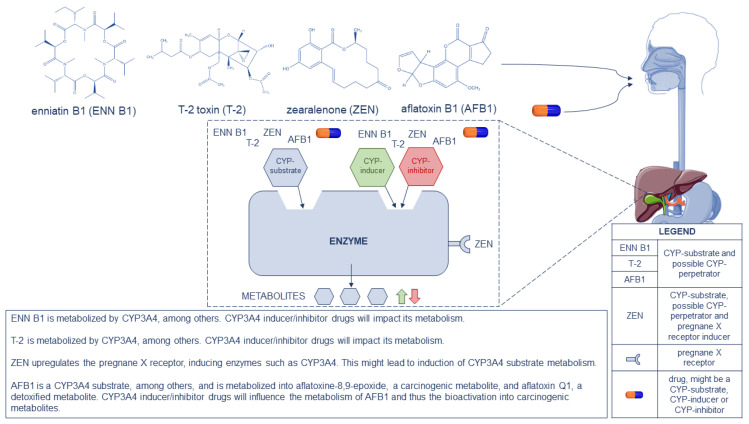
Representation of enniatin B1 (ENN B1), T-2 toxin (T-2), zearalenone (ZEN), aflatoxin B1 (AFB1) and drugs as CYP450 substrates and potential CYP450 perpetrators, following the gastrointestinal tract to the liver where metabolism occurs. A schematic drawing is given on the possible pathways that might take place in the human liver. In the bottom-right corner, a legend is displayed. At the bottom-center an explanation is given for ENN B1, T-2, ZEN and AFB1, as CYP3A4 substrates and as pregnane X receptor inducer in case of ZEN, on the possible impact on metabolism.

**Table 1 toxins-14-00873-t001:** Overview of a selection of mycotoxins, associated IARC classification and toxic effects based on animal models, human cell lines or human case–control studies. * IARC = International Agency for Research on Cancer. Group 1 = carcinogenic to humans. Group 2B = possibly carcinogenic to humans. Group 3 = not classifiable as to its carcinogenicity to humans.

Mycotoxin	IARC Classification *	Toxic Effects
Aflatoxin B1	Group 1 [11]	Cancer, hepatotoxicity, immunosuppression [12]
Fumonisin B1	Group 2B [11]	Cancer, hepatotoxicity, leuko-encephalomalacia, teratogenic effects [12,13]
Zearalenone	Group 3 [14]	Infertility, abortion, cervical and breast cancer [15]
Deoxynivalenol	Group 3 [14]	Gastrointestinal toxicity, inflammation of central nervous system [16]
Ochratoxin A	Group 2B [14]	Nephrotoxicity, cancer, teratogenic and immunotoxic effects [17,18]
T-2-toxin	Group 3 [14]	Dermatitis, diarrhea, hemorrhages, necrosis of bone marrow, spleen, ovary, testis and gastrointestinal lining. [19]

**Table 2 toxins-14-00873-t002:** Summary of the main CYP-enzymes involved in the human metabolism of the listed mycotoxins.

Mycotoxin	Involved CYP-Enzymes	References
AFB1	CYP3A4, CYP3A5, CYP3A7, CYP1A2, CYP2A13	[55,56,57,58]
DON	No phase I metabolism	[59,60,61,62,63]
T-2	CYP3A4, CYP2E1, CYP1A2, CYP2B6, CYP2D6 and CYP2C19	[64,65]
ZEN	CYP3A4, CYP3A5, CYP2C8, CYP1A2	[66,67]
FB1	No phase I metabolism	[68,69,70,71]
ENN B1	CYP3A4, CYP3A5, CYP1A2, CYP2C19	[72,73]

**Table 3 toxins-14-00873-t003:** Summary of the impact of mycotoxins on CYP450 enzymes and receptors. The summary comprises data of in vitro and in vivo animal experiments, across species.

Mycotoxin	Induction	Inhibition	In Vitro/In Vivo	References
aflatoxin B1	CYP1A2 CYP2B6 CYP2C9 CYP3A4 CYP3A5	CYP1A1 (rabbits) CYP3A6 (rabbits)	In vitro	[76,78]
deoxynivalenol	CYP2B	/	In vitro (mice)	[76,79]
T-2 toxin	/	CYP1A1 CYP1A2 CYP2A1 CYP2B4 CYP1A4 CYP1A5 CYP3A37	In vivo (rabbits) In vivo (chicken)	[80] [81]
zearalenone	PXR-receptor	/	In vitro	[77]
fumonisin B1	CYP1A4 (chicken) CYP2E1 (rats)	CYP1A2 (rats) CYP2C11 (rats)	In vivo	[76,82,83]
enniatin B1	/	CYP2C19	In vitro	[84]

**Table 4 toxins-14-00873-t004:** Overview of reported interactions between exogenous compounds, such as drugs, and mycotoxins, in the literature. Per reported interaction it is indicated which compound was involved as a perpetrator and as a victim.

Perpetrator	Victim	References
antifungal drugs with azole structure	mycotoxins with an azole structure	[74]
butylated hydroxytoluene	aflatoxin B1	[86]
oltipraz	aflatoxin B1	[87,88,89,90]
zearalenone	midazolam	[91] *
T-2 toxin	midazolam	[91] *

* only statistically relevant for the elimination constants (K_e_), indicating that further research is necessary.

## Data Availability

Not applicable.

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
