# Peer review of "Possible Mechanisms of the Interplay between Drugs and Mycotoxins—Is There a Possible Impact?"

_toxins, 2022, doi:10.3390/toxins14120873_

Round 1

Reviewer 1 Report

This is a well-aimed review about the possible mycotoxin-drug interactions with potential public health implications. The topic is unique and focused. I have only one recommendation to improve the manuscript's overall quality:

-          Authors should design a schematic drawing about the interactions of mycotoxins with CYP450 enzymes (chapter 3.2) and/or about the proved mycotoxin-drug interactions (chapter 4) instead of the quite trivial competitive – non-competitive inhibtion types. The latter is a basic biological knowledge presented and illustrated by hundreds of educational books already, so it does not really add any value to the manuscript.

- I especially liked the "Physiologically-Based Pharmacokinetic Modelling To Predict Mycotoxin-Drug Interactions" chapter, because it describe a unique issue and an innovative method. Are there really just 3 references to this chapter? Are there no other endavours or ongoing trials in this topic?

Reviewer 2 Report

This paper is a review that deals with mycotoxin contamination and its interaction with other xenobiotics.  The authors described several types of these interactions with some detail. However, I believe the paper can gain scientific value if considering some revisions as follows:

The abstract should provide information about the findings.  The current version just talks about what to find in the paper, but not the actual results of the search. 

The authors spent a lot of effort on toxin metabolism, with little explanation on the importance of this biochemical activation.

Figure 3,1 is totally irrelevant, please remove it. Instead, the authors should include a Figure showing the structures of most known mycotoxins and those xenobiotics frequently reported to interact with them.

The question stated in the tittle is not answered in the manuscript. Is there a possible impact?  The authors should emphasize on answering this question, both in the Abstract and the Conclusions.

Avoid using abbreviations when starting a paragraph.

The review will be a lot better if known mycotoxin-xenobiotic interactions are initially described in Tables.  These may also include the number of publications reported for each pair.

I think the possible mechanisms involved in the mycotoxin-xenobiotic interactions were not well described.  A figure showing them would clarify the manuscript.

The authors should also include the main molecular mechanisms involved in these interactions, in particular the type of receptor involved and the possible role of bioactivation on such interaction.

The conclusions need much more work. Please use what you found and conclude from that. New research is also needed, but something has to be concluded with the available information.

Round 2

Reviewer 2 Report

The paper has been improved, but still it needs some minor changes.

You have many "impacts" in the abstract. Change them.  

Rephrase this sentence: The only statistically-sound research on this matter reports mycotoxins as a victim and drugs as perpetrators in interactions. This looks as a result, not a conclusion. 

Figure 3.1 is terrible. Please increase clarity. All words should read fine.
